# Implications in Cancer of Nuclear Micro RNAs, Long Non-Coding RNAs, and Circular RNAs Bound by PRC2 and FUS

**DOI:** 10.3390/cancers16050868

**Published:** 2024-02-21

**Authors:** Guruprasadh Swaminathan, Diana G. Rogel-Ayala, Amine Armich, Guillermo Barreto

**Affiliations:** 1Université de Lorraine, CNRS, Laboratoire IMoPA, UMR 7365, F-54000 Nancy, France; 2Lung Cancer Epigenetics, Max-Planck-Institute for Heart and Lung Research, 61231 Bad Nauheim, Germany

**Keywords:** miRNA, lncRNA, circRNA, PRC2, EZH2, FUS, cancer

## Abstract

**Simple Summary:**

In this review we summarize recent articles on the role of specific biotypes of non-coding RNAs in different cancer types. We will focus on micro RNAs that has been characterized in the cell nucleus, as well as long non-coding RNAs and circular RNAs that are bound by components of the polycomb repressive complex 2 or by the protein Fused in Sarcoma. We will describe mechanistic aspects offering comprehensive insights into specific diagnostic and therapeutic implications of the non-coding RNAs in the cancer types described in the review.

**Abstract:**

The eukaryotic genome is mainly transcribed into non-coding RNAs (ncRNAs), including different RNA biotypes, such as micro RNAs (miRNAs), long non-coding RNAs (lncRNAs), and circular RNAs (circRNAs), among others. Although miRNAs are assumed to act primarily in the cytosol, mature miRNAs have been reported and functionally characterized in the nuclei of different cells. Further, lncRNAs are important regulators of different biological processes in the cell nucleus as part of different ribonucleoprotein complexes. CircRNAs constitute a relatively less-characterized RNA biotype that has a circular structure as result of a back-splicing process. However, circRNAs have recently attracted attention in different scientific fields due to their involvement in various biological processes and pathologies. In this review, we will summarize recent studies that link to cancer miRNAs that have been functionally characterized in the cell nucleus, as well as lncRNAs and circRNAs that are bound by core components of the polycomb repressive complex 2 (PRC2) or the protein fused in sarcoma (FUS), highlighting mechanistic aspects and their diagnostic and therapeutic potential.

## 1. Introduction

The genome in a eukaryotic cell is mainly transcribed into RNAs that are not translated into proteins. These so-called non-coding RNAs (ncRNAs) include long non-coding RNAs (lncRNAs; >200 nucleotides long) and micro RNAs (miRNAs; 21–25 nucleotides long) [1]. LncRNAs are important regulators of different biological processes in the nucleus [2,3]. Together with other factors, lncRNAs provide a framework for the assembly of defined chromatin structures at specific loci, thereby silencing repetitive DNA elements, modulating centromere function, and regulating gene expression [2,3,4,5]. On the other hand, miRNAs are assumed to act primarily in the cytosol by inhibiting translation [6,7]. Currently, the miRBase database (www.mirbase.org; accessed in 1 December 2023) lists over 2588 distinct miRNAs that are involved in nearly every cellular process, highlighting the vast and complex nature of this field of study [7,8]. Moreover, alterations in miRNA expression have been implicated in a myriad of diseases, emphasizing their importance in maintaining cellular homeostasis [9,10,11,12,13,14,15,16,17].

The biogenesis of miRNAs begins with the transcription of their genes by RNA polymerase II, generating primary miRNAs (pri-miRNAs). These pri-miRNAs are then processed in the nucleus by the DROSHA-DGCR8 complex into preliminary miRNAs (pre-miRNAs) [18], which are transported from the nucleus to the cytosol by Exportin-5 in a Ran-GTP-dependent manner [19]. In the cytosol, the pre-miRNAs undergo further cleavage by the enzyme DICER, resulting in a small double-stranded RNA [20,21]. One strand of this double-stranded RNA, determined by its thermodynamic properties, is then incorporated into the RNA-induced silencing complex (RISC), where it guides the complex to target messenger RNAs (mRNAs) to reduce gene expression, either through mRNA degradation or inhibition of translation [22]. Even though miRNAs are supposed to act mainly in the cytosol [6,7], mature miRNAs have been also detected in the nuclei of different cells [23,24,25,26,27,28]. Moreover, a hexanucleotide element has been reported to direct miRNA nuclear import [29]. However, the function of miRNAs in the cell nucleus has barely been investigated. In this review, we will summarize the cancer-related findings of miRNAs that have been detected in the cell nucleus.

We have demonstrated in previous studies that a nuclear miRNA, lethal 7d (*Mirlet7d*, also known as *let-7d*), is part of the multicomponent ncRNA–protein complex MiCEE, which mediates epigenetic repression of bidirectionally transcribed genes [23,30]. In this context, nuclear *Mirlet7d* binds ncRNAs expressed from these bidirectionally transcribed genes. The miRNA-lncRNA duplexes are bound by the exosome-associated protein C1D, which in turn serves as a dock for the RNA-degrading exosome complex [31,32,33] and the polycomb repressive complex 2 (PRC2) [34,35]. While the exosome RNA-degrading complex mediates degradation of the ncRNA, PRC2 induces heterochromatin at the promoter of the gene, thereby reducing the transcription of the protein-coding RNA. The name of the MiCEE complex is due to its components, the nuclear *Mirlet7d*, C1D, the nuclear-specific exosome subunit EXOSC10, and the histone methyl transferase EZH2 from PRC2 (*Mirlet7d*-C1D-EXOSC10-EZH2). Whereas the function of the exosome RNA-degrading complex is well documented in virtually all aspects of RNA biology [31,32,33], the role of the RNA-binding activity of various components of PRC2 is less documented [36,37]. The PRC2 core consists of four subunits: EZH2 (enhancer of zeste homolog 2) or its closely related homolog EZH1, SUZ12 (suppressor of zeste 12), EED (embryonic ectoderm development), and RBBP4 (retinoblastoma binding protein 4) [35]. We will discuss in the present review the RNA-binding activity of the subunits of the PRC2 core within the context of different cancer types.

The human FUS protein family is made up of more than 25 proteins [3,38]. The first member of this protein family was FUS [39]. In addition to FUS, EWSR1 and TAF15, known together as FET proteins, TDP-43, and hnRNPA1 have been reported to mediate liquid-liquid phase separation [40]. Common characteristics of FET proteins are long, intrinsically disordered domains; domains with repeated motifs; and domains for interaction with other proteins, RNA, or DNA [41,42]. FUS is an RNA-binding protein involved in various aspects of RNA metabolism. In addition, FUS and different translocations of FUS have been related to various pathologies, including amyotrophic lateral sclerosis [43], frontotemporal lobar degeneration [44], and liposarcoma [39], among others. We have unpublished results demonstrating the functional interaction between FUS and various components of the MiCEE complex during transcriptional regulation and three-dimensional (3D) genome organization. Motivated by our findings, we included in this review recent publications focusing on the RNA-binding activity of FUS, which has been related to different cancer types.

## 2. Nuclear miRNAs and Their Implications in Cancer

Nuclear miRNAs have been identified and profiled across various cell types, including different cancer cell lines, through advanced techniques such as next-generation sequencing (NGS), RNA fluorescence in situ hybridization (FISH), Northern blot, and quantitative polymerase chain reaction after reverse transcription (qRT-PCR), showing their localization not just to the cytoplasm but also to the nucleus [23,24,25,26,27,28,29]. These discoveries broaden the understanding of miRNA functions beyond post-transcriptional regulation in the cytoplasm to include the regulation of processes in the cell nucleus, such as chromatin structure, transcriptional control, and RNA processing, among others [45]. This is further evidenced by the study on the HCT116 colon cancer cell line, where mature miRNAs were found within the nucleus, hinting at nuclear functions beyond the established cytoplasmic post-transcriptional regulation [24]. These combined insights suggest a revised understanding of miRNA function, indicating their broader regulatory potential within cellular biology.

The human miRNA 29b (*miR-29b*) is predominantly localized in the cell nucleus [29]. This study suggests that *miR-29b* participates in the regulation of transcription or the splicing of target transcripts within the cell nucleus, rather than the miRNA-canonical translation regulatory functions in the cytosol. The nuclear localization of *miR-29b* is directed by a distinctive hexanucleotide 3′-terminal motif with the nucleotide sequence 5′-AGUGUU-3′ (Figure 1, top), which is not present in *miR-29a*. Remarkably, the hexanucleotide 3′-terminal motif of *miR-29b* acts as a transferable nuclear localization element that directs nuclear enrichment of other miRNAs or small interfering RNAs to which it is attached. Moreover, we found by sequence alignment analysis that a partially conserved and extended version of this nuclear shuttling motif is present in mouse and human miRNAs that have been functionally characterized in the cell nucleus [46]. The presence of a nuclear shuttling motif in various miRNAs, which in turn is conserved across species, suggests the existence of a common mechanism regulating the enrichment of these miRNAs in the cell nucleus, thereby indicating that even miRNAs with common seed sequences can have diverse functions and regulatory capacities due to distinct subcellular localizations, underscoring the complexity of miRNA-mediated regulation.

The function of *miR-29b* has been reported in various cancer types [49]. The specific role of *miR-29b* in melanoma has been shown by targeting transcripts like *LAMC1* and *LASP1* to reduce cell invasiveness, which could be crucial for stopping disease progression [50] (Figure 2, left top). In colorectal cancer, *miR-29b* overexpression is associated with suppression of the ERK/EGFR signaling pathway, leading to apoptosis and a decrease in angiogenesis and metastasis [51]. Furthermore, in acute myeloid leukemia, *miR-29b* reduces the levels of human antigen R (HuR, also known as ELAVL1 for embryonic lethal abnormal vision 1) by targeting its transcript [52], resulting in decreased levels of RELA (v-rel avian reticuloendotheliosis viral oncogene homolog A, also known as p65 or NFKB3) in the cell nucleus and reduced phosphorylation levels of RELA, IκBα, STAT1, STAT3, and STAT5. All these proteins are components of the NF-κB and JAK/STAT signaling pathways. The inhibition of these key oncogenic pathways mediated by *miR-29b* is accompanied by cell cycle arrest, cell viability decrease, apoptosis increase, and reduction of invasion and migration. These findings highlight the multifaceted roles of *miR-29b* in cancer, where its nuclear regulatory functions could be intrinsically linked to its ability to control key processes implicated in oncogenesis and cancer progression (Figure 2, left).

The miRNA lethal 7 (*Mirlet7*, also known as *let-7*) was together with *lin-4* one of the first miRNAs discovered. *Mirlet7* was initially discovered in *Caenorhabditis elegans*, where it plays a critical role in proliferation and differentiation of stem cells [59]. Subsequently, *MIRLET7* was the first miRNA identified in humans and its sequence is highly conserved across species. In vertebrates, the *Mirlet7* family consists of 10 miRNAs, including *Mirlet7a*, *Mirlet7b*, *Mirlet7c*, *Mirlet7d*, *Mirlet7e*, *Mirlet7f*, *Mirlet7g*, *Mirlet7i*, *miR-98*, *miR-202*, which are encoded by 13 genes, underlining the genetic complexity and significance of this miRNA family. In addition to its sequences, the functions of the *Mirlet7* family members seem to be conserved across species, since they are integral to various biological processes such as development, stem cell biology, aging, and metabolism [60,61]. Reduced levels of *Mirlet7* family members have been related to less-differentiated cellular stages and to different cancers [60,61]. Interestingly, LIN28B and its paralog, LIN28A, interact with the primary and/or preliminary transcripts of various members of the *Mirlet7* family during development and oncogenesis and suppress the biogenesis of the mature miRNAs [62,63,64,65,66]. LIN28A together with Musashi1 (MSI1) sequester *pri-Mirlet7* in the cell nucleus [66]. In addition, LIN28A also binds *pre-Mirlet7* in the cytoplasm to block processing by DICER and recruits TUT4/7 (terminal uridylyltransferase 4/7), which adds uridyl groups to the 3′ end of *pre-Mirlet7*, thereby tagging it for degradation [67,68,69,70]. In contrast, LIN28B sequesters *pri-Mirlet7* transcripts in the nucleolus, preventing the biogenesis of mature *Mirlet7* [71]. As commented above, mature *Mirlet7d* participates as part of the MiCEE complex during epigenetic silencing of bidirectionally transcribed genes (Figure 1, left). In addition, *Mirlet7d* is involved in the structure and function of the nucleolus, the cellular hub for ribosomal RNA synthesis and ribosome biogenesis [23]. By tethering specific genomic regions to the nucleolus periphery, *Mirlet7d* contributes to the establishment of perinucleolar heterochromatin-associated domains (PNHADs), thus actively participating in the dynamic interface between 3D genome organization and gene expression regulation [23]. The role of mature *Mirlet7d* in nucleolar organization complements and adds another layer of complexity to our understanding of the functions of *Mirlet7* family members [61,72]. The integrated view of all these molecular mechanisms underscores the multifaceted nature of non-coding RNAs and their regulatory proteins in both normal cellular processes and disease states. Recent studies have revealed that reduced *Mirlet7d* levels in oral squamous cell carcinoma are linked to enhanced epithelial-mesenchymal transition (EMT) and increased migratory and invasive capabilities of cancer cells [53]. Additionally, *Mirlet7d* downregulation correlated with heightened chemoresistance in oral squamous cell carcinoma, underlining its crucial role in cancer progression and treatment response (Figure 2, left bottom). Conversely, increased *Mirlet7d* levels reversed EMT, reducing invasiveness and enhancing chemosensitivity in cancer cells. In breast cancer, reduced levels of *Mirlet7d* have been related to enhanced cancer hallmarks [73,74]. On the other hand, the lncRNA *MAFG-AS1* (MAF BZIP transcription factor G antisense RNA 1) promotes proliferation and metastasis of breast cancer by different mechanisms (Figure 2, right top), such as modulating the STC2 (Stanniocalcin 2) pathway and activating the JAK2/STAT3 (Janus kinase 2/signal transducer and activator of transcription 3) pathway through the *miR-3196*/TFAP2A axis [54,55]. Interestingly, we have shown that *Mirlet7d* induced degradation of *MAFG-AS1* in the cell nucleus through the MiCEE complex [23]. Moreover, we have unpublished results demonstrating that *Mirlet7d* reduced cancer hallmarks in breast cancer cells by degradation of *MAFG-AS1* and suppression of the JAK2/STAT2 signaling pathway. All these reports together suggest the development of therapies for oral squamous cell carcinoma and breast cancer using *Mirlet7d*.

The function of *miR-9* in the cell nucleus illustrates different aspects of nuclear miRNA [27]. It has been reported that *miR-9* exerts regulatory functions within the nucleus by targeting the lncRNA *MALAT1* (metastasis-associated lung adenocarcinoma transcript 1), which plays a role in cancer progression (Figure 1, right). The authors demonstrated that *miR-9* in the cell nucleus binds directly to *MALAT1* via specific binding motifs, leading to its degradation in an AGO2 (Argonaute 2)-dependent manner. The degradation of *MALAT1* mediated by *miR-9* was shown in osteosarcoma cells MG-63 that were treated with 17β-estradiol, which increased *miR-9* levels, degraded *MALAT1*, and reduced several cancer hallmarks, including cell proliferation, colony formation, migration, and invasion (Figure 2, right) [57]. These findings not only shed light on the intricate regulatory mechanisms involving ncRNAs in the nucleus but also suggest a broader functional landscape for *miR-9* in gene regulation beyond its traditional cytoplasmic activities.

Highlighting the role of *miR-9* in the cell nucleus, we recently published a sequencing experiment after chromatin isolation by miRNA purification (ChIRP-seq) in mouse lung fibroblasts showing that *miR-9* is enriched at promoters and super-enhancers of genes that are responsive to tissue growth factor beta 1 (TGFB1) signaling [46]. Further, we found that nuclear *miR-9* is required for chromatin features related to increased transcriptional activity, including broad domains of the euchromatin histone mark histone 3 tri-methylated lysine 4 (H3K4me3) and the non-canonical DNA secondary structures G-quadruplex (G4). Moreover, we showed by chromosome conformation capture-based methods that nuclear *miR-9* is required for promoter-super-enhancer looping. Our study places nuclear *miR-9* in the same functional context as G4s and promoter-enhancer interactions during 3D genome organization and transcriptional activation induced by TGFB1 signaling, a critical regulator of proliferation programs in cancer [75].

A recent study underscores the function of *miR-9* as a tumor suppressor in acute myeloid leukemia [56]. The authors found that *miR-9* and *CXCR4* (C-X-C chemokine receptor 4) were differentially expressed in acute myeloid leukemia samples, showing an inverse correlation between *miR-9* and *CXCR4* (Figure 2, right). The acute myeloid leukemia patients with high levels of *CXCR4* and low expressions of *miR-9* showed poor prognosis. Further, *miR-9* overexpression inhibited proliferation, apoptosis resistance, migration, and invasion of acute myeloid leukemia cells. The authors also demonstrated that *miR-9* directly targets *CXCR4*, with the *miR-9*/*CXCR4* axis emerging as a critical factor and potential therapeutic target in acute myeloid leukemia pathology.

Another miRNA that has been functionally characterized in the cell nucleus is *miR-126-5p* [47]. Santovito et al. reported that nuclear *miR-126-5p* inhibits caspase-3 and confers endothelial protection by autophagy in atherosclerosis (Figure 1, right bottom). Mechanistically, *miR-126-5p*, in association with AGO2 and Mex3a (Mex-3 RNA-b Binding Family Member A), is transported to the nucleus via autophagic vesicles. In the cell nucleus, the authors showed that *miR-126-5p* dissociates from Ago2 and binds to caspase-3 with its seed sequence, preventing dimerization of the caspase and inhibiting its activity to block apoptosis. The non-canonical inhibition of caspase-3 by nuclear *miR-126-5p* is a mechanism by which miRNAs can modulate protein function.

Hepatocellular carcinoma is one of the most common gastrointestinal malignancies, with the third-highest mortality rate [76]. In the context of this cancer type, it has been reported that *miR-126-5p* promotes tumor cell proliferation, metastasis, and invasion by targeting tryptophan 2,3-dioxygenase (TDO2), which is a key enzyme in the tryptophan–kynurenine metabolic pathway (Figure 2, right bottom) [58]. By RNA immunoprecipitation (RIP) using TDO2-specific antibodies and sequencing of the precipitated RNAs, the authors identified 645 known miRNAs and 138 novel miRNAs that can bind to TDO2 in human hepatocellular carcinoma (HCCLM3) cells. By gain-of-function experiments after transfection of *miR-126-5p* mimics in HCCLM3 cells, the authors detected significantly increased expression levels of *PI3K*, *AKT*, *WNTI1*, and *CTNNB1* (β-catenin). Loss-of-function experiments after transfection of *miR-126-5p* inhibitors in HCCLM3 cells reduced the expression levels of these transcripts. These results suggested that the PI3K/AKT pathway and the WNT pathway may be involved in *miR-126-5p*-related promotion of proliferation and metastasis. Although this study supports that the interaction between TDO2 and *miR-126-5p* plays a role in hepatocellular carcinoma, it is not clear whether the observed effects take place in specific cellular compartments due to limitations in the experimental systems that were used. For example, RIP was performed with whole-cell lysate without cellular fractionation. In ovarian cancer cells, the expression of *miR-126-5p* was reported to be high and to correlate with increased proliferation, migration, and invasion, while inhibiting apoptosis [77]. Mechanistically, *miR-126-5p* targets the *PTEN* transcript in the cytosol, thereby activating the PI3K/Akt/mTOR pathway. Further, the RNA methyltransferase-like 3 (METTL3) promotes *miR-126-5p* maturation through N6-methyladenosine (m6A) modification in the cell nucleus, thereby enhancing its activity. Silencing METTL3 disrupts this mechanism, offering therapeutic potential against ovarian cancer by targeting *miR-126-5p* and its regulatory mechanisms.

In line with the roles of miRNAs in the cell nucleus, *miR-584-3p* has been shown to participate in transcription regulation of the gene coding for matrix metalloproteinase 14 (*MMP-14*) in the nucleus of gastric cancer cells [48]. The transcription factor Yin Yang 1 (YY1) directly binds the promoter and mediates the transcription of the *MMP-14* gene (Figure 1, left bottom). On the other hand, the authors found that *miR-584-3p* binds to sites that are adjacent to the YY1 binding site at the *MMP-14* promoter, interacts with AGO2, and recruits EZH2 and euchromatic histone lysine methyltransferase 2 (EHMT2). The formation of this RNA–protein complex results in enrichment of repressive epigenetic markers at the *MMP-14* promoter, decreased binding of YY1, and reduced *MMP-14* transcription, thereby inhibiting the tumorigenesis and aggressiveness of gastric cancer.

## 3. Non-Coding RNAs Bound by PRC2 and Their Implications in Cancer

In addition to nuclear *Mirlet7d* as a component of the MiCEE complex, we will discuss the RNA-binding activity of other components of this ncRNA–protein complex, namely the subunits of the PRC2 core. The recruitment of PRC2 to specific genomic sites is a pertinent question that needs to be answered. A growing body of literature supports that different RNA molecules interact in vivo with PRC2 proteins [78,79,80,81]. The interaction of PRC2 components with specific RNA molecules contributes to the recruitment of the whole complex to specific genomic loci to regulate chromatin structure and transcription. For example, it has been reported that PRC2 members interact with nascent RNA at transcriptionally active genes, which determines the recruitment of PRC2 components to chromatin [80]. EZH2 is the enzymatic catalytic subunit of PRC2 that can alter downstream target gene expression by trimethylation of Lys-27 in histone 3 (H3K27me3). In addition, EZH2 can regulate gene expression in other ways besides H3K27me3 [82,83,84]. Earlier studies have revealed that EZH2 possesses an RNA-binding domain in the amino acid residues 342–368, and the phosphorylation of specific residues is cell cycle regulated and increases the binding to ncRNAs [85]. Further, it has been reported that promiscuous RNA binding by PRC2 members depends on the length of the RNA, where shorter RNAs are bound with lower affinity [86]. The same group proposed that RNA binding by EZH2 is not random but selective to maintain the repressed state of chromatin of those target genes that have escaped repression.

EZH2 is actively involved in transcription regulation of genes that are crucial for various biological mechanisms including cell fate, cell cycle, cell differentiation, DNA damage repair, lineage specification, autophagy, apoptosis, and tumorigenesis [87,88,89]. Further, studies show the involvement of EZH2 in the regulation of tumor microenvironment and antitumor immune response in solid cancers, directly affecting immunotherapy efficacy [90,91,92,93]. Notably, in a clinical study involving 696 patients, high EZH2 levels were positively correlated with increased tumor cell proliferation in four major cancer types including cutaneous melanoma and cancers of the endometrium, prostate, and breast [94]. Interestingly, the authors further demonstrated that high EZH2 levels led to poor survival in all four cancer types studied in a population-based setting. This study suggests that high EZH2 levels can be used as a predictive factor to identify increased tumor cell proliferation and aggressive subgroups in several cancers, and that it may be used as target for the development of therapies. The regulatory effects of ncRNAs binding to EZH2 have significant therapeutic potential in different cancers. However, they have received less attention. Seminal contributions focusing on ncRNAs related to EZH2 in major cancers are discussed below (see also Figure 3, top).

It has been reported that EZH2 is highly expressed in various cancer types [95,96,97] and mediates epigenetic silencing by catalyzing the heterochromatin histone mark H3K27me3 at specific loci, which in turn induces changes in gene expression leading to abnormal biological functions [82,83,84,98]. Increasing evidence has reported that the lncRNA ATB (*lnc-ATB*) promoted tumor progression in breast [99], prostate [100], and colon cancers [101], therefore suggesting *lnc-ATB* as a potential prognostic marker and therapeutic target in human cancers [102]. Chen et al. have reported that levels of *lnc-ATB* were increased in ovarian cancer tissue and high levels of *lnc-ATB* were associated with poor outcomes of patients with ovarian cancer [103]. Further, the authors demonstrated that *lnc-ATB* directly interacts with EZH2 by RIP and RNA pull-down assays in human ovarian cancer cell lines SKOV3 and A2780. Interestingly, *lnc-ATB* loss-of-function in SKOV3 and A2780 cells reduced cancer hallmarks, such as cell proliferation, invasion, and migration. In addition, chromatin immunoprecipitation (ChIP) assay after *lnc-ATB* loss-of-function demonstrated that *lnc-ATB* is required for enrichment of EZH2 and heterochromatic histone mark H3K27me3 at promoters of tumor suppressor genes, such as *CDX1*, *FOXC1*, *LATS2*, *CDH1*, and *DAB2IP*. Accordingly, the expression of these tumor suppressor genes was increased after *lnc-ATB* loss-of-function. These results suggest *lnc-ATB* as a potential biomarker for ovarian cancer diagnosis.

Neuroblastoma is a common extracranial solid tumor type with poor prognosis in children. The lncRNA *MEG3* (maternally expressed 3) is found to be downregulated in neuroblastoma [104]. *MEG3* overexpression in neuroblastoma cell lines attenuated autophagy through inhibition of FOXO1 and EMT via the mTOR pathway. Moreover, in a second article the authors showed that *MEG3* and EZH2 negatively regulate each other [105]. RIP and co-immunoprecipitation (Co-IP) experiments were used to demonstrate the interaction of *MEG3* and EZH2. Further, *MEG3* exerted anti-cancer effects by mediating ubiquitination of EZH2, thereby leading to its degradation. Conversely, EZH2 interacted with DNMT1 and HDAC1 to induce silencing of *MEG3*. Summarizing both articles, reduced *MEG3* levels and increased EZH2 levels form a feedback loop that promotes the development of neuroblastoma. Combined blockage of EZH2 and HDAC1 with the appropriate inhibitors may be an effective treatment strategy for neuroblastoma cases with low *MEG3* and high EZH2 levels. Recently, a novel therapeutic approach using polymer nanoparticle-based administration of the lncRNA *MEG3* was designed to control the progression of hepatocellular carcinoma in HepG2 cells [106].

It has been reported that the lncRNA pseudogene misato family member 2 (*MSTO2P*) affects cell proliferation, apoptosis, metastasis and invasion in hepatocellular carcinoma through the PI3K/AKT/mTOR pathway [107], thereby supporting the diagnostic and prognostic value of *MSTO2P*. A recent study explored the link between EZH2-mediated epigenetic silencing and ncRNAs, specifically the lncRNA *MSTO2P* in colorectal cancer [108]. *MSTO2P* levels were high in colorectal cancer tissues and cells. Further, experiments performed in colon cancer cells (HT-29 and SW480) revealed that loss-of-function of *MSTO2P* suppressed cell proliferation and invasion, promoting cell cycle arrest and apoptosis. In addition, the group also demonstrated that *MSTO2P* interacts with EZH2 in the cell nucleus and mediates epigenetic silencing of the tumor suppressor gene cyclin-dependent kinase inhibitor 1A (*CDKN1A*).

The lncRNA *SNHG22* (small nucleolar RNA host gene 22) is highly expressed in gastric cancer cells and tissues and is correlated with poor prognosis in patients with gastric cancer [109]. The authors of this study found that the transcription factor ELK4 (ETS transcription factor ELK4) binds to the promoter region of the *SNHG22* and promotes its expression in gastric cancer cells. Further, RNA pull-down using biotinylated *SNHG22* and nuclear protein extracts from gastric cancer cells followed by mass spectrometry analysis of enriched proteins showed that *SNHG22* directly interacts with EZH2. Consistently, RIP using EZH2-specific antibodies showed specific enrichment of *SNHG22*. Further, ChIP assays in BGC-823 and MGC-803 cells showed that EZH2 and H3K27me3 are enriched at promoters of various tumor suppressor genes (*CDH1*, *EAF2*, *ADRB2*, *RUNX3*, and *RAP1GAP*) in a *SNHG22*-dependent manner, since *SNHG22* loss-of-function reduced the levels of EZH2 and H3K27me3 and increased the expression of all tumor suppressor genes analyzed, which in turn also inhibited the proliferation and invasion ability of gastric cancer cells. Another study reported that *SNHG22* induces invasion, migration, and angiogenesis via the *miR-361-3p*/HMGA1/Wnt axis in gastric cancer cells [109]. Therefore, targeting the biomarker candidate *SNHG22* could be a promising strategy for the diagnosis and prognosis of gastric cancer.

A newly identified lncRNA named UPK1A antisense RNA 1 (*UPK1A-AS1*) was found to promote cellular proliferation and tumor growth by accelerating cell cycle progression in hepatocellular carcinoma cells [110]. Moreover, cell cycle-related genes including *CCND1*, *CDK2*, *CDK4*, *CCNB1*, and *CCNB2* were observed to be upregulated in hepatocellular carcinoma cells overexpressing *UPK1A-AS1*. Mechanistically, *UPK1A-AS1* directly interacted with EZH2 to mediate its nuclear translocation, therefore leading to increased H3K27me3 levels. The authors reported that targeting EZH2 with specific small interfering RNA impaired *UPK1A-AS1*-mediated upregulation of cell cycle-related genes. Additionally, *UPK1A-AS1* was significantly increased in hepatocellular carcinoma tissue, and increased levels of *UPK1A-AS1* in hepatocellular carcinoma patients predicted poor prognosis. Thus, *UPK1A-AS1* exhibits potential as a novel biomarker for the prognosis and therapy of hepatocellular carcinoma.

In a clinical study involving 89 patients with non-small cell lung cancer, the expression of the lncRNA taurine-upregulated gene 1 (*TUG1*) was determined to be significantly downregulated [111]. Using *TUG1* 4C sequencing and bioinformatic analysis, the authors found *CELF1* (CUGBP and Elav-like family member 1) to be a potential target of *TUG1* in-trans regulation. Subsequent experiments by RIP showed that *TUG1* was bound to two components of the PRC2 core, namely EZH2 and EED. Moreover, ChIP assays showed that EZH2 and EDD were enriched at the promotor of the *CELF1* gene in a *TUG1*-dependent manner. The study showed that *TUG1* could bind to PRC2 at the promoter region of *CELF1* and negatively regulate *CELF1* expressions in lung squamous cell carcinoma H520 cells. These results may facilitate the development of new treatment modalities targeting TUG1/PRC2/CELF1 interactions in patients with lung squamous cell carcinoma.

Elevated expression of the lncRNA tyrosine kinase non receptor 2 antisense RNA 1 (*TNK2-AS1*) is reported in acute myeloid leukemia cell lines and is negatively correlated with patients’ survival [112]. The authors found that *TNK2-AS1* exerted tumor-promoting activity in acute myeloid leukemia cells. Mechanistically, increased levels of *TNK2-AS1* in acute myeloid leukemia cells were mediated by transcription factor ELK1 (ETS domain-containing protein-1). Further, EZH2 bound to *TNK2-AS1* silenced the gene coding for *CELF2* (CUGBP Elav-like family member 2) and exerted tumor-promoting effects through activation of the PI3K/Akt pathway. In contrast, knockdown of *TNK2-AS1* markedly reduced cell proliferation and promoted apoptosis and differentiation in acute myeloid leukemia cells, suggesting *TNK2-AS1* as a potential therapeutic target and prognostic marker for patients with acute myeloid leukemia. Another study reported the positive feedback loop between STAT3 and *TNK2-AS1*-mediated dysregulated STAT3 signaling by elevating VEGFA expression to facilitate angiogenesis in non-small cell lung cancer, indicating *TNK2-AS1* as a potential target for therapeutic intervention [113].

EZH2-mediated epigenetic silencing of tumor suppressor genes leading to cancer proliferation has been reported earlier [114]. The transcriptional programs controlled by EZH2 were investigated in normal germinal center B cells of lymphomagenesis [115]. This study uncovered EZH2 binding sites at approximately 1800 promoter regions and identified key cell cycle-related tumor suppressor genes that are specifically downregulated in germinal center B cells of lymphomagenesis. In addition, the involvement of various long and short non-coding RNAs modulating the function of EZH2 in different cancer types was comprehensively reviewed by Mirzaei et al. [116].

In addition to EZH2, other PRC2 core proteins that have RNA-binding activity are SUZ12 and EED [80]. Recent literature that reports RNA-binding activity of SUZ12 and EED implicated in cancers is summarized below (see also Figure 3, bottom).

It has been reported that the lncRNA *MARCKSL1-2* (MARCKSL1-transcript variant 2, NR_052852.1) promoted tumor progression by regulating epithelial–mesenchymal transition [117], thereby supporting the prognostic and therapeutic value of this lncRNA. In addition, the function of *MARCKSL1-2* in docetaxel resistance of lung adenocarcinoma cells was investigated [118]. The authors found that SUZ12 is recruited by *MARCKSL1-2* to the promoter of *HDAC1* and increases H3K27me3 levels, thereby reducing *HDAC1* expression. The reduced levels of HDAC1 blocked the suppressive effect of HDAC1 on histone acetylation modification at the *miR-200b* promoter, which in turn resulted in increased *miR-200b* expression. In summary, *MARCKSL1-2* increases *miR-200b* expression by repressing *HDAC1* expression. This hypothesis was further validated by in vivo experiments using a mouse xenograft tumor model that supported *MARCKSL1-2* overexpression leading to attenuated docetaxel resistance in lung adenocarcinoma tumors.

The expression of *miR-767-5p* was identified to be significantly upregulated in glioblastoma tissues and cell lines. The authors demonstrated SUZ12 as a putative target of *miR-767-5p,* and the possibility that *miR-767-5p* acts by regulating SUZ12 expression [119]. It was revealed that the inhibitory effects of *miR-767-5p* on glioblastoma cell phenotypes were reversed by overexpression of SUZ12, indicating that the forced upregulation of *miR-767-5p* may represent a novel therapeutic strategy for glioma patients by targeting SUZ12. In vitro experiments suggest that the ectopic expression of *miR-767-5p* led to reduced proliferation, promoting cell cycle arrest and apoptosis in glioblastoma cell lines and inhibiting glioblastoma tumor growth in a mouse xenograft model.

Hepatocellular carcinoma is regarded as one of the most common malignancies leading to cancer-related death worldwide. The function of the lncRNA *RBM5-AS1* (RBM5 antisense RNA 1) in the development of hepatocellular carcinoma was studied [120]. *RBM5-AS1* levels are high in hepatocellular carcinoma tissues and cell lines, especially in Hep3B and HepG2 cells. Further, *RBM5-AS1* loss-of-function reduced cell proliferation, invasion, and migration of hepatocellular carcinoma cells. Mechanistically, *RBM5-AS1* was shown to recruit PRC2 core components (EZH2, SUZ12, EED) to the *miR-132/212* promoter, elevate H3K27me3 levels, and reduce *miR-132/212* expression. In summary, *RBM5-AS1* was shown to act together with PRC2 as epigenetic regulator by repressing *miR-132/212* expressions, thereby promoting hepatocellular carcinoma progression.

## 4. Non-Coding RNAs Bound by FUS and Their Implications in Cancer

Various lncRNAs have been associated with the severity and prognosis of hepatocellular carcinoma [121]. Among these transcripts are *MALAT1*, *HOTAIR*, *HOTTIP*, *H19*, and *GTL2* (also known as *MEG3*) (Figure 4, top). A metadata study looking for RNA-binding proteins (RBP) that are potentially binding these lncRNAs identified FUS together with eIF4AIII and PTB as the proteins binding to the majority of the previously identified lncRNAs [122]. In addition, we have unpublished results demonstrating the functional interaction between FUS and components of the MiCEE complex during transcriptional regulation and 3D genome organization. Even though the mechanism by which FUS affects the development of hepatocellular carcinoma is unknown, the data suggest that FUS might play a role in the pathology by interacting with the identified lncRNAs.

The lncRNA *SLC8A1-AS1* was shown to be downregulated in papillary thyroid cancer clinical samples. Moreover, the overexpression of *SLC8A1-AS1* led to reduced proliferation and increased apoptosis in papillary thyroid cancer cells. A recent study demonstrated that the observed effect is due to *SLC8A1-AS1* binding FUS as well as the mRNA of NUMB-like endocytic adaptor protein (*Numbl*) [123]. Furthermore, a loss-of-function of FUS showed similar results as the *SLC8A1-AS1* downregulation, leading to decreased levels of *Numbl* and increased cell proliferation, even when overexpression of *SLC8A1-AS1* was induced, suggesting that FUS is a key part of the mechanism. This lncRNA-FUS-mRNA interaction leads to stability and the maintained expression of *Numbl*, which in turn is an inhibitor of the Notch signaling pathway. The Notch pathway regulates cell proliferation and differentiation, and it has been reported to be hyperactive in several human cancers. Therefore, the evidence suggesting that *SLC8A1-AS1*-FUS interaction stabilizes *Numbl* expression, a Notch inhibitor, fits with the observations of decreased proliferation and increased apoptosis [124].

In a recent metadata study performed to identify lncRNAs associated with prostate cancer, *LMNTD2-AS1* was found to be overexpressed. Increased levels of this transcript correlated with poor overall survival as well as poor progression-free interval in the TCGA database [125]. Furthermore, the levels of *LMNTD2-AS1* correlated with the outcome of primary therapy. For example, patients with low levels of *LMNTD2-AS1* showed a complete positive response to the therapy, whereas patients with high levels of *LMNTD2-AS1* showed partial response to the therapy. Interestingly, *LMNTD2-AS1* loss-of-function experiments showed a significant reduction in the proliferation, migration, and invasion of prostate cancer cells. *LMNTD2-AS1* is reported to be bound by several RBPs, for example, FUS, CSTF2, SMNDC1, and RANGAP1. Strikingly, FUS is the only RBP protein also showing strong correlation with poor overall survival and progression-free interval in the TCGA database. Further, FUS gain-of-function experiments showed an increase in the proliferation of prostate cancer cells, rescuing the effect observed by the *LMNTD2-AS1* loss-of-function.

Non-coding RNAs and RNA-binding proteins can affect cancer progression not only by influencing cell migration and proliferation but also by promoting drug resistance. High levels of the lncRNA *LINC01133* have been associated with resistance to treatment with sorafenib in patients with pancreatic cancer [126]. The mechanism describes *LINC01133* increasing the resistance of pancreatic cancer cells to ferroptosis. This process occurs by *LINC01133* stabilizing FSP1 mRNA. Such stabilization is achieved through the binding of FUS, forming a *LINC01133*-FUS-*FSP1* complex.

FUS has also been associated with tumor suppressor roles. High levels of the lncRNA *SOX2OT* have been correlated with poor prognosis in patients with pancreatic ductal adenocarcinoma [127]. The authors found that *SOX2OT* directly binds to FUS implementing ChIRP and RIP assays that were performed in in pancreatic ductal adenocarcinoma cells. In another manuscript, the authors showed that the binding of *SOX2OT* to FUS promoted the ubiquitination and degradation of FUS [128]. Downstream, FUS regulates cell cycle-associated factors, such as CCND1 and p27. Thus, the degradation of FUS mediated by *SOX2OT* leads to cell cycle disruption. The *SOX2OT*–FUS regulatory axis promotes migration, invasion, tumor growth, and metastasic abilities of pancreatic cancer cells. Pancreatic cancer is an extremely aggressive cancer type. Only 8% of patients with pancreatic cancer are alive five years after the disease is diagnosed. Therefore, the urge to find effective therapeutic target is pressing. A better understanding of the epigenetic mechanism of the disease would offer the opportunity to develop novel therapeutic approaches.

Circular RNA (circRNA) is a biotype of single-stranded RNA that shows a circular structure, a product of a back-splicing process. CircRNAs have gained importance in recent years since they have been reported to participate in the regulation of several biological processes [129]. *CircRNA_0000285* is highly expressed in cervical cancer tissues when compared to control tissue [130]. Proliferation and migration of cervical cancer cells were significantly reduced after *circRNA_0000285* loss-of-function experiments. Likewise, FUS levels decreased after *circRNA_0000285* loss-of-function. Further results indicated that the expression level of FUS was positively correlated with the expression of *circRNA_0000285* in cervical cancer tissues. In addition, the knockdown of *circRNA_0000285* significantly inhibited the formation and metastasis of cervical cancer in nude mice. In summary, *circRNA_0000285* was able to enhance the proliferation and metastasis of cervical cancer by increasing FUS levels (Figure 4, bottom), which might be a potential therapeutic target for cervical cancer treatment.

Interactions between FUS and circRNA have been reported to be involved in metastatic processes of other cancer types. For example, *circROBO1* was found to be upregulated in liver metastasis cells derived from breast cancer [131]. The overexpression of *circROBO1* was associated with a high rate of proliferation and migration in breast cancer cells. Loss-of-function of *circROBO1* showed a decrease in both proliferation and migration of breast cancer cells. In contrast, gain-of-function of *circROBO1* promoted liver metastasis and tumor growth in murine models. In addition, *circROBO1* increased KLF5 levels by sponging *miR-217-5p*. In turn, KLF5 regulates the expression of FUS. Interestingly, FUS is involved in the back-splicing process of *circROBO1*, thereby establishing a positive feedback loop. Thus, the *circROBO1*/KLF5/FUS axis becomes both, a potential biomarker and a therapeutic target for breast cancer metastatic cells. In a similar case, *circEZH2* was found to be upregulated in liver metastasic cells derived from breast cancer [132]. The overexpression of *circEZH2* is associated with poor prognosis in breast cancer. In addition, overexpression of *circEZH2* promoted in vivo liver metastasis in murine models. In vitro loss-of-function experiments targeting *circEZH2* showed decreased cell migration and invasion. Similar to *circROBO1*, *circEZH2* also absorbs *miR-217-5p*, as well leading to increased KLF5 levels and FUS transcription activation. FUS is also involved in the back-splicing of *circEZH2*, therefore showing a positive feedback loop.

CircRNAs have also been associated with tumor suppressor roles. In colorectal cancer, tumor-suppressive circRNAs are selectively secreted in exosomes to maintain cancer cell viability [133]. However, while the levels of *circSKA3* were significantly high in colorectal cancer tissue, it was not found in serum from patients with colorectal cancer. *CircSKA3* was retained in colorectal cancer cells via a specific intra-cellular motif, which the authors called cellmotif element. Furthermore, the zinc-finger transcription factor snail 2 (SNAI2, also known as SLUG) bound *circSKA3* by the cellmotif element, which stabilized SNAI2 by inhibiting its ubiquitination and degradation [134]. SNAI2 has already been reported to promote EMT in patients with colorectal cancer. Interestingly, FUS plays a key role in the back-splicing and circularization of *circSKA3*, binding also to the cellmotif element of *circSKA3*.

## 5. Conclusions and Future Directions

The discovery of mature miRNAs in the cell nucleus opens an intriguing chapter in cancer research, underscoring the complexity and versatility of miRNA functions. The presence of miRNAs like *miR-29b*, *Mirlet7d*, *miR-9*, *miR-126-5p*, and *miR-584-3p* in the nucleus and their diverse regulatory roles, ranging from direct gene silencing to influencing intricate signaling pathways in cancer, signify a paradigm shift in our understanding of miRNA biology. These findings challenge the traditional view of miRNAs as predominantly cytoplasmic entities and highlight their potential in nuclear processes such as transcriptional regulation, epigenetic modification, and chromatin organization.

Clinical studies have shown that high levels of the core components of PRC2 correlated with increased cancer hallmarks and the poor survival of patients with various cancer types [94]. Thus, high levels of PRC2 core components can be used as a predictive factor to identify increased tumor cell proliferation and aggressive subgroups in several cancers, and they may be used as target for the development of therapies. Mechanistically, in most of the studies presented here, PRC2 mediates the heterochromatic histone mark H3K27me3 at promoters of tumor suppressor genes. Further, the interaction of PRC2 core components with specific RNA molecules contributes to the recruitment of the whole complex to specific genomic loci to regulate chromatin structure and transcription. The regulatory effects of ncRNA binding on PRC2 core components, especially on EZH2 as the enzymatic catalytic subunit of the complex, have significant therapeutic potential in different cancers.

The studies presented here suggest that the role played by FUS in specific cancer types may depend on the RNAs with which it is interacting in the cancer cells. Interestingly, FUS interacts with several circRNAs within the context of different cancer types, which may be a result of the role that FUS plays during the back-splicing process of specific circRNAs. The regulatory effects of circRNAs on the function of FUS and other FET proteins may have potential for the development of therapies in various cancer types.

## Figures and Tables

**Figure 1 cancers-16-00868-f001:**
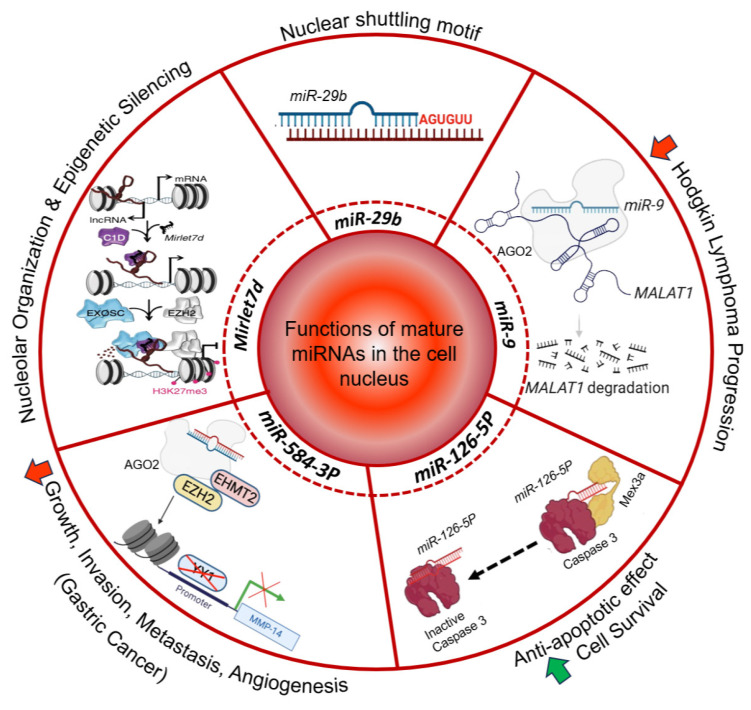
Functions of mature miRNAs in the cell nucleus. The nuclear shuttling motif of *miR-29b* is highlighted [29]. The diagram further indicates the participation of *Mirlet7d* in nucleolar organization and epigenetic silencing [23]. *miR-9* is associated with a decrease in Hodgkin lymphoma progression [27]. *miR-126-5p* is linked to apoptotic control and cell survival [47]. *miR-584-3p* is linked to gastric cancer [48]. Red arrow, decrease; green arrow, increase.

**Figure 2 cancers-16-00868-f002:**
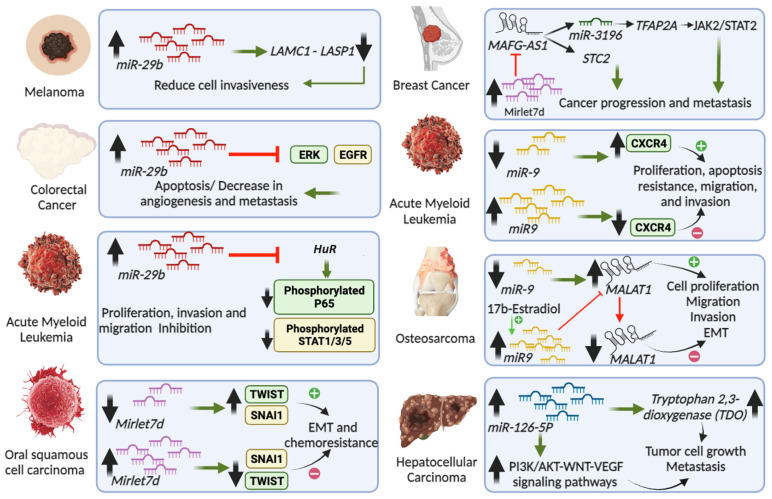
Role in cancers of miRNAs that have been characterized in the cell nucleus. The figure depicts miRNAs’ involvement in various cancer types. For example, *miR-29b* in melanoma [50], colorectal cancer [51], and acute myeloid leukemia [52]; *MIRLET7D* in oral squamous cell carcinoma [53] and breast cancer [23,54,55]; *miR-9* in acute myeloid leukemia [56] and osteosarcoma [57]; and *miR-126-5p* in hepatocellular carcinoma [58].

**Figure 3 cancers-16-00868-f003:**
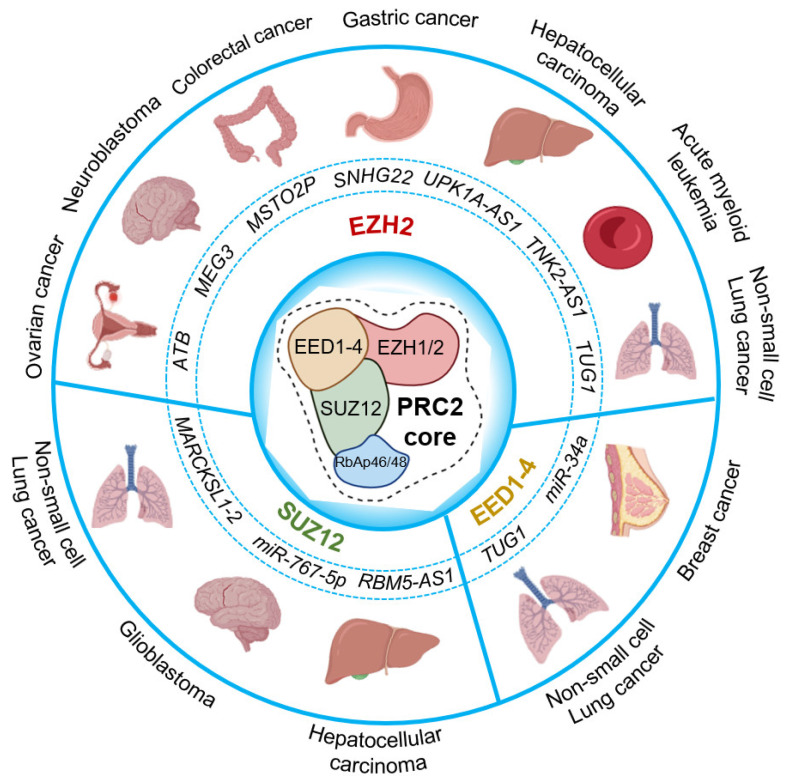
Cancer-related RNA-binding activity of PRC2 core components. The circle chart above illustrates several long non-coding RNAs and micro-RNAs that are reported to bind the core components of PRC2, implicated in different cancer types. PRC2 core in the center; non-coding RNAs that bind to EZH2 (top); SUZ12 (bottom left); and EED1-4 (bottom right) are represented.

**Figure 4 cancers-16-00868-f004:**
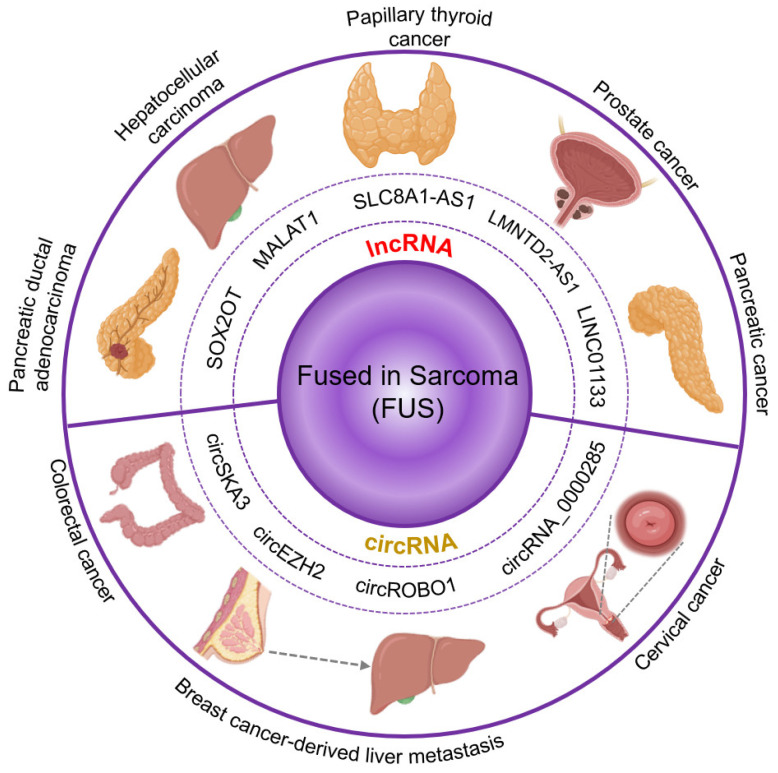
Cancer-related RNA-binding activity of fused in sarcoma. FUS has been related to several types of cancer by binding to different biotypes of RNA. In the lncRNAs already reported to have an effect in cancer that also interact with FUS, we found *SOX2OT,* associated with pancreatic ductal adenocarcinoma. *MALAT1*, associated with hepatocellular carcinoma. *SCL8A1-AS1*, associated with papillary thyroid cancer. *LMNTD2-AS1*, associated with prostate cancer. *LINC01133*, associated with pancreatic cancer. Another RNA biotype we found to be frequently bound by FUS and also associated with cancer was circular RNA. Among them, we found *circSKA3*, associated with colorectal cancer; *circRNAs_0000285* associated with cervical cancer; and *circEZH2* and *circROBO1* associated with breast cancer-derived liver metastasis.

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
