# Peer review of "Implications in Cancer of Nuclear Micro RNAs, Long Non-Coding RNAs, and Circular RNAs Bound by PRC2 and FUS"

_cancers, 2024, doi:10.3390/cancers16050868_

Round 1
Reviewer 1 Report
Comments and Suggestions for Authors
The manuscript "Implications in Cancer of Nuclear Micro RNAs, Long Non-Coding RNAs, and Circular RNAs Bound by PRC2 and FUS" provides a comprehensive overview of the roles of various non-coding RNAs (ncRNAs) in cancer, particularly focusing on their interactions with polycomb repressive complex 2 (PRC2) and FUS protein. The authors have successfully compiled and summarized recent studies in this evolving field, highlighting the importance of ncRNAs in nuclear processes such as transcriptional regulation, chromatin organization, and epigenetic modifications in cancer.
The manuscript's strength lies in its detailed review of nuclear miRNAs and their roles in cancer, providing novel insights into the nuclear functions of miRNAs beyond traditional cytoplasmic post-transcriptional regulation. The discussion on the involvement of miRNAs, lncRNAs, and circRNAs in regulatory complexes like PRC2 and their association with cancer pathogenesis is thorough and informative. This section effectively illustrates the complexity of ncRNA-mediated regulation in cancer biology. Additionally, while the authors have covered a wide range of studies, future directions or potential therapeutic implications of these findings in cancer treatment are not extensively discussed. A dedicated section outlining potential therapeutic strategies targeting these ncRNAs or their associated complexes could add significant value to the manuscript.
Overall, this manuscript is a valuable contribution to the field of cancer biology, particularly in understanding the nuclear roles of ncRNAs. It opens up avenues for further research into the therapeutic potential of targeting these molecules in cancer treatment.
Reviewer 2 Report
Comments and Suggestions for Authors
The article is at an acceptable level in terms of quality and comprehensiveness. It is suggested that it be accepted by improving the quality of the article in terms of grammar.
Comments on the Quality of English LanguageThe article is at an acceptable level in terms of quality and comprehensiveness. It is suggested that it be accepted by improving the quality of the article in terms of grammar.
Reviewer 3 Report
Comments and Suggestions for Authors
Dear Authors,
The manuscript distinctly delineates the role of specific classes of ncRNAs (miRNAs, lncRNAs, circRNAs) in the context of cancer. Additionally, the objectives of the study, such as mechanisms of action and diagnostic/therapeutic potential, are aptly presented.
The manuscript delves into various classes of ncRNAs, ranging from miRNAs conventionally associated with the cytosol to lncRNAs and circRNAs possessing nuclear functionalities. This contributes to a broader comprehension of the intricacies of gene regulation in the context of cancer.
Significantly, the paper investigates the associations of miRNAs, lncRNAs, and circRNAs with key molecular actors such as PRC2 and FUS. This indicates a potentially pivotal role for these elements in the regulation of the cancer process.
Recommendations for enhancement:
Expansion on diagnostic/therapeutic potential: Offer more comprehensive insights into specific diagnostic and therapeutic implications wherever feasible. This can aid in shaping conclusions and suggest potential applications of this knowledge in medicine.
Language and Style: Strive for a clear, precise, and concise writing style. Avoid excessive use of technical terms without adequate explanation.
Conclusion: Integrate a clear conclusion that succinctly summarizes the key points of the study and potentially points towards future research or therapeutic implications.
In sum, the manuscript holds the potential to be informative and beneficial in understanding the role of ncRNAs in cancer. However, additional clarity in structure may enhance its overall strength.
Best regards.
Reviewer 4 Report
Comments and Suggestions for Authors
The submitted manuscript summarises the current knowledge on the roles of non-coding RNAs in the nucleus and binding to EZH2 and FUS in cancers. The topic is interesting and potentially influential, but I feel that the manuscript could be improved in several points.
1. My main concern is that there is little apparent coherence in the manuscript, the three different parts are not forming a logical sequence and there is little information given about the rationale behind selecting and combining these three specific topics. I encourage the authors to provide a more conscious reasoning for their choice of subjects and create a more comprehensive line of thought throughout the manuscript.
2. Section 1, which is about the role of nuclear miRNAs, is somewhat problematic in terms of scientific quality. The main message would be that nuclear miRNAs have different roles in cellular life than the classic cytoplasmic function of mRNA regulation. Yet, in several cases the introduced mechanisms for the miRNAs is exactly that and it is not always clarified if the actual function is connected to the nuclear localisation of the miRNA. The most striking example is miR-126-5p (lines 229-247), where there is no reference provided for its role in hepatocellular carcinoma. I assume the authors referenced a paper with the title of "MiR-126-5p Promotes Tumor Cell Proliferation, Metastasis and Invasion by Targeting TDO2 in Hepatocellular Carcinoma" (DOI: 10.3390/molecules27020443). However, in this article there is no indication whether the original authors have separated nuclear and cytoplasmic fractions when checking for the interacting partners. In reference 73, the original authors show that the effect of miR-126-5p on PTEN is in fact a result of the canonical, mRNA regulatory pathway and the nuclear events only involve the m6A methylation of the miR. I suggest the authors make clearer distinction between the proven nuclear effects of miRNAs and the functions that are related to their canonical role.
3. There are some redundancies in the text (mainly when introducing different cancer types, but also about the various roles of ncRNAs), which could be reduced.
Minor points:
1. I suggest to include key citations for each nuclear miRNAs in the legend of Figure 2. The legend needs updating too, as it provides more detailed description for the first couple of examples than the rest. There is no strong need for the detailed descriptions, as they are given in the text, but if the authors decide to keep them, than they should be provided for all examples.
2. Figure numbers are accidentally included on some figures - they should be removed.
Comments on the Quality of English Language
Moderate English corrections are needed as there are some minor inaccuracies in the language usage and grammar.
Round 2
Reviewer 4 Report
Comments and Suggestions for Authors
I thank the authors for considering my suggestions, they responded to all my criticisms and the paper can be published in its current form.